# Generation of Vaccine Candidate Strains That Antigenically Match Classical Swine Fever Virus Field Strains

**DOI:** 10.3390/vaccines13020188

**Published:** 2025-02-14

**Authors:** Maya Kobayashi, Loc Tan Huynh, Saho Ogino, Lim Yik Hew, Miki Koyasu, Hikaru Kamata, Takahiro Hiono, Norikazu Isoda, Yoshihiro Sakoda

**Affiliations:** 1Laboratory of Microbiology, Department of Disease Control, Faculty of Veterinary Medicine, Hokkaido University, Sapporo 060-0818, Japantanloc@ctu.edu.vn (L.T.H.); hiono@vetmed.hokudai.ac.jp (T.H.); nisoda@vetmed.hokudai.ac.jp (N.I.); 2Faculty of Veterinary Medicine, College of Agriculture, Can Tho University, Can Tho 900000, Vietnam; 3Gifu Central Livestock Hygiene Service Center, Gifu 501-1112, Japan; 4Gifu Hida Livestock Hygiene Service Center, Gifu 506-8688, Japan; 5One Health Research Center, Hokkaido University, Sapporo 060-0818, Japan; 6International Collaboration Unit, International Institute for Zoonosis Control, Hokkaido University, Sapporo 001-0020, Japan; 7Hokkaido University Institute for Vaccine Research and Development (HU-IVReD), Hokkaido University, Sapporo 001-0021, Japan

**Keywords:** antigenicity, classical swine fever virus, chimeric vaccine, E2 envelope glycoprotein, pestivirus

## Abstract

Background: Classical swine fever virus (CSFV) is genetically categorized into three genotypes. A live-attenuated vaccine strain GPE^−^, currently used in Japan, belongs to genotype 1 and is genetically distinct from the field strains circulating in Japan, which belong to genotype 2. This study aimed to understand the antigenicity of recent field isolates in Japan and develop new vaccine candidates that antigenically match field strains. Methods: The serum samples of 20 pigs vaccinated with GPE^−^ were subjected to a serum neutralizing test (SNT) using one of the field strains, CSFV/wb/Jpn-Mie/P96/2019 (Mie/2019). For the antigenic matching, vGPE^−^/HiBiT/Mie E2 was generated by replacing the viral glycoprotein E2, the main target of the neutralizing antibody, with that of Mie/2019. Additionally, vGPE^−^/HiBiT/Mie E2/PAPeV E^rns^ was generated by further substituting glycoprotein E^rns^ with that of pronghorn antelope pestivirus (PAPeV) since E^rns^ is not important as a vaccine immunogen and can be replaced by that of other pestiviruses to provide an immunological marker. The efficacy of vGPE^−^/HiBiT/Mie E2/PAPeV E^rns^ was further evaluated by the challenge experiments in pigs. Results: The SNT titers of serum sample against Mie/2019 were 6.1-fold lower than that against vGPE^−^. The generated recombinant viruses showed closer antigenicity to Mie/2019 than vGPE^−^. The challenge study confirmed that vGPE^−^/HiBiT/Mie E2/PAPeV E^rns^ provided clinical and virological protection against a field CSFV equivalent to vGPE^−^. Conclusions: This study demonstrated that swapping the E2 encoding region with the prevalent field CSFVs is a promising strategy to achieve antigenic matching between the vaccine and field strains.

## 1. Introduction

Classical swine fever (CSF) is a fatal contagious disease of domestic pigs and wild boars. Because CSF causes significant economic losses, it is one of the important viral diseases in pigs and a disease notifiable to the World Organisation for Animal Health (WOAH) [1,2]. CSF virus (CSFV) is a causative agent of CSF, belonging to the *Pestivirus* genus within the *Flaviviridae* family. Its viral genome is a single-stranded positive-sense RNA of ~12.3 kb long, with a single large open reading frame (ORF), encoding four structural proteins (C, E^rns^, E1, and E2) and eight nonstructural proteins (N^pro^, p7, NS2, NS3, NS4A, NS4B, NS5A, and NS5B) [3].

CSFV strains are categorized into three major genotypes (1–3) and further divided into several subgenotypes (1.1–1.4, 2.1–2.3, and 3.1–3.4) based on the complete E2 sequence [4]. The GPE^−^, CSFV that adapted to guinea pig cells and does not exhibit exaltation of Newcastle disease virus phenomenon is a live-attenuated vaccine strain currently used for domestic pigs in Japan. This strain belongs to genotype 1. In contrast, the field strains circulating in Japan since 2018 belong to genotype 2 [4,5]. However, there is only one CSFV serotype; the antigenic gap between the strains of genotypes 1 and 2 has been recognized [6]. The potential antigenic mismatch could lead to a lower efficacy of vaccination against prevalent field CSFVs, especially in piglets, as they are typically protected from CSFVs only by maternal antibodies. Therefore, due to a gradual decrease in maternal antibody levels before the appropriate vaccination timing, piglets may have an insufficient immune status against antigenically distinct field strains, resulting in a high risk of CSFV infection. This finding partially supports the fact that 35 CSF cases were reported even in vaccinated farms in Japan [7,8] after the CSF outbreak was reported in Gifu Prefecture in September 2018 [4,5]. Thus, the antigenic variations of currently prevalent CSFVs should be analyzed to understand the potential risk of CSF outbreaks and update vaccine strains.

The WOAH stipulates a CSF-free country and zone in which neither CSF cases have been recognized nor vaccination against CSF has been implemented in domestic and captive wild pigs unless there are means of distinguishing between vaccinated and infected pigs [8,9]. In Japan, except for Hokkaido Prefecture, an island prefecture physically separated from the main island of Japan, the vaccination program for domestic pigs with GPE^−^ strain had been implemented in addition to biosecurity management as preventive measures. To achieve CSF-free status again under the vaccination program, pigs infected with field CSFVs must be differentiated from vaccinated ones [differentiating infected from vaccinated animals (DIVA)]. To develop DIVA vaccines based on vGPE^−^, an infectious cDNA clone of GPE^−^, vGPE^−^/PAPeV E^rns^ and vGPE^−^/PhoPeV E^rns^ were generated by replacing the E^rns^ region of vGPE^−^ with those of the atypical pestiviruses, pronghorn antelope pestivirus (PAPeV) and phocoena pestivirus, which are genetically distinct from CSFVs [10]. These two chimeric viruses conferred sufficient vaccine efficacy to 2-week-old piglets equivalent to vGPE^−^ [10]. Meanwhile, specific antibodies against CSFV E^rns^ were not detected from pig antisera immunized with each recombinant virus. This proves the successful installation of the DIVA property to the vaccine strain, vGPE^−^ [11].

Because most CSFVs are non-cytopathogenic in infected cells, immunological staining is routinely performed to detect viral antigens, although it is time- and labor-intensive. Recently, the NanoLuc binary technology (NanoBiT) [12] was utilized to generate recombinant CSFV carrying a small reporter tag [13]. NanoBiT consists of a high-affinity NanoBiT (HiBiT) and a large NanoBiT and works as a reporter gene by forming a heterodimer. In creating new CSFV vaccine candidate strains, HiBiT consisting of 11 amino acids was employed as an inserted tag based on its usefulness in measuring the virus infectivity titer and antibodies by the serum neutralization test (SNT).

In this study, the antigenicity of a current field strain in Japan, CSFV/wb/Jpn-Mie/P96/2019 (Mie/2019), was compared to that of vGPE^−^. Two novel CSF vaccine candidates, vGPE^−^/HiBiT/Mie E2 and vGPE^−^/HiBiT/Mie E2/PAPeV E^rns^, were generated by the replacement of E2 with that of Mie/2019 and E^rns^ with that of PAPeV. SNT analyzed the antigenicity of these new recombinant viruses. These viruses were experimentally infected in pigs to assess their pathogenicity. Because vGPE^−^/HiBiT/Mie E2/PAPeV E^rns^ possesses a potential immunological marker, it was used for further in vivo study. Finally, the vaccine efficacy of vGPE^−^/HiBiT/Mie E2/PAPeV E^rns^ against Mie/2019 and its DIVA property were evaluated through animal experiments.

## 2. Materials and Methods

### 2.1. Cells and Viruses

We established swine kidney line-L (SK-L) cells and maintained them with Eagle’s minimum essential medium (EMEM; Shimadzu Diagnostics Corp., Tokyo, Japan). This medium was supplemented with 0.3 mg/mL L-glutamine (Nacalai Tesque, Inc., Kyoto, Japan), 100 U/mg penicillin G (Meiji Seika Pharma Co., Ltd., Tokyo, Japan), 8 µg/mL gentamicin (TAKATA Pharmaceutical Co., Ltd., Saitama, Japan), sodium bicarbonate (Nacalai Tesque), 0.1 mg/mL streptomycin (Meiji Seika Pharma), 0.295% tryptose phosphate broth (Becton, Dickinson and Company, Franklin Lakes, NJ, USA), 10 mM *N,N*-bis-(2-hydroxyethyl)-2-aminoethanesulfonic acid (Merck KGaA, Darmstadt, Germany), and 10% horse serum (Thermo Fisher Scientific, Inc., Waltham, MA, USA). SK-L cells were incubated at 37 °C in the presence of 5% CO_2_ for the propagation and titration of CSFV.

The field strain Mie/2019 (accession no. LC713055), which was isolated from a wild boar in 2019 in Mie Prefecture, Japan, was used in this study. In addition, two recombinant clones of CSFVs were used: a recombinant clone derived from a CSF live-attenuated vaccine, vGPE^−^ [14], and a HiBiT luciferase gene-encoding vGPE^−^, vGPE^−^/HiBiT [13]. The recombinant viruses were rescued using the corresponding plasmid-based cDNA clones, namely pGPE^−^ and pGPE^−^/HiBiT.

### 2.2. Virus Titration by Immunoperoxidase Staining (IPX) and Luciferase Assay

SK-L cells were infected with 10-fold serial dilutions of CSFVs in 96-well plates (Thermo Fisher Scientific). After the incubation at 37 °C for 4 days, cells were fixed at 80 °C for 1 h and immunostained with a primary monoclonal antibody against the NS3 protein as described previously [14]. Cells were incubated at room temperature (RT) for 1 h. Cells were washed with phosphate-buffered saline (PBS) and incubated with goat anti-mouse IgG (H + L) horseradish peroxidase conjugate (Bio-Rad Laboratories) at RT for 1 h. Viral antigens were detected as red signal stained with 3-amino-9-ethyl carbazole (Merck), and virus titers were calculated and expressed as the 50% tissue culture infectious dose (TCID_50_)/mL [15].

For titration of HiBiT-fused viruses, SK-L cells were infected with 10-fold serial dilutions of CSFVs in 96-well plates (Thermo Fisher Scientific) and incubated at 37 °C. The cell culture supernatant was collected after 4 days of incubation to measure luminescence using a Nano-Glo HiBiT lytic detection system (Promega Corp., Madison, WI, USA) according to the manufacturer’s manual. An equal volume of culture supernatant and Nano-Glo HiBiT lytic buffer was mixed in Nunc™ F96 MicroWell™ White Polystyrene Plate (Thermo Fisher Scientific) and kept at RT for 10 min. The luminescence of this mixture (40 µL) was measured using a Synergy H1 Microplate Reader (Agilent Technologies, Inc., Santa Clara, CA, USA). In this study, the luminescence cutoff value was set to 20.

### 2.3. Serum Samples

The field serum samples collected from 20 pigs vaccinated with the commercial GPE^−^ vaccine were provided by a pig farm in Mie Prefecture, Japan. In this farm, the vaccine was administered at 30–40 days of age, and serum samples were collected at 100–120 days of age. Pig antisera against GPE^−^ was produced and provided by the National Institute of Animal Health, National Agriculture and Food Research Organization, Japan. Antiserum against vGPE^−^/HiBiT/Mie E2 was prepared from two 2-week-old crossbred Landrace × Duroc × Yorkshire SPF pigs (Yamanaka Chikusan, Hokkaido, Japan). Pigs were inoculated intramuscularly with 10^7.0^ TCID_50_ of vGPE^−^/HiBiT/Mie E2. Sera were collected at 35 days postinoculation (dpi).

### 2.4. SNT

The serum samples were inactivated at 56 °C for 30 min. Each equal volume of serially diluted serum and 100 TCID_50_ of CSFVs were mixed and incubated at 37 °C for 1 h. The mixture and SK-L cell suspension were incubated in 96-well plates (Thermo Fisher Scientific) at 37 °C with 5% CO_2_. Viruses were detected at 4 dpi by IPX, as described above.

In the antigenicity assessment, each serum sample was tested with four wells per virus, and its neutralizing antibody titer was indicated as the reciprocal of the highest serum dilution that prevents virus growth in 50% of four replicate wells as a 50% neutralizing dose (ND_50_). The titers were calculated according to the Kärber equation [16].

In the experimental infection study, ND_50_ was indicated as the reciprocal of the highest serum dilution that prevented virus growth in 50% of two replicate wells, according to the WOAH manual [8].

### 2.5. Plasmid Construction and Virus Rescue

pGPE^−^/HiBiT/Mie E2 was constructed from the cDNA of E2 of Mie/2019 and the backbone of pGPE^−^/HiBiT. The E2 insert and vector fragments were amplified using PrimeSTAR GXL DNA Polymerase (TaKaRa Bio, Inc., Shiga, Japan). The specific primers for this amplification are indicated in Appendix A. These fragments were fused using the In-Fusion HD cloning Kit (TaKaRa Bio) according to the procedures indicated in Figure 1. The pGPE^−^/HiBiT/Mie E2/PAPeV E^rns^ was constructed from the cDNA of E^rns^ of PAPeV [10] and the backbone of pGPE^−^/HiBiT/Mie E2, according to the same protocol as mentioned above (Figure 1). The nucleotide sequences of the plasmids were confirmed as described below.

CSFV clones were rescued from the plasmids carrying full-length CSFV cDNA under the control of the T7 promoter. The plasmid was linearized at an *Srf*I site at the end of the viral genomic cDNA sequence. The purified DNA by phenol-chloroform extraction and ethanol precipitation was used for run-off transcription using the MEGAscript T7 kit (Thermo Fisher Scientific). After the purification of the DNase I-digested sample using MicroSpin S-400 HR columns (Cytiva, Tokyo, Japan), RNA was electroporated to SK-L cells using Gene Pulser Xcell (Bio-Rad Laboratories, Inc., Hercules, CA, USA), set at 200 V and 500 µF. After the following incubation at 37 °C for 3 days, the virus was recovered, and the infectivity titer was confirmed as described below.

### 2.6. Genetic Stability Assessment

The genetic stability of the recombinant viruses was evaluated over five blind passages in SK-L cells. The culture supernatants (100 μL) were inoculated into naïve SK-L cells. At 72 h after inoculation, the culture supernatants were collected and inoculated into newly prepared naïve SK-L cells. The virus titers of each passage were determined in triplicate using SK-L cells according to the previously described method. The viral genome sequencing was also conducted using the cell culture supernatants collected at each passage.

### 2.7. Sequencing

Nucleotide sequencing for the full-length cDNA clones of each recombinant virus and the entire genomes of the rescued viruses were performed. The BigDye Terminator version 3.1 Cycle Sequencing Kit (Thermo Fisher Scientific) and an ABI 3500 Genetic Analyzer (Thermo Fisher Scientific) were used for the sequencing of cDNA clones and PCR fragments from viral RNA. Sequencing data were analyzed using GENETYX^®^ Network Edition version 15.0.1 (Nihon Server Corp., Tokyo, Japan).

### 2.8. Viral Growth Kinetics and Bioassay for Type I Interferon (IFN) Measurement

The growth kinetics of vGPE^−^/HiBiT, vGPE^−^/HiBiT/Mie E2, and vGPE^−^/HiBiT/Mie E2/PAPeV E^rns^ in SK-L cells were assessed by inoculating them into confluent cell monolayers at a multiplicity of infection (MOI) of 0.001. After inoculation, SK-L cells were incubated at 37 °C with 5% CO_2_. Cell supernatants were collected daily from 0 to 7 dpi for the virus titration and type I IFN measurement. According to the abovementioned method, viral titers of each sample were determined in triplicate in SK-L cells.

Swine IFN-α/β was measured following previously established methods [17]. In summary, an ultraviolet cross-linker (DNA-FIX DF-254; ATTO, Tokyo, Japan) was used for the supernatants of SK-L cells infected with each virus to inactivate infectious virus before being inoculated into SK6-MxLuc cells carrying an Mx/Luc reporter gene [17]. The recombinant swine IFN-α produced according to the previous study was used as the standard. After 20 h incubation, cell lysates were prepared by adding 100 µL passive lysis buffer to the SK6-MxLuc cells. The firefly luciferase activities were measured, and type I IFN was quantified using the Dual-Luciferase Reporter Assay System and POWERSCAN^®^4 (Agilent Technologies International Japan Ltd., Tokyo, Japan). Results were documented across three independent experiments. Each type I IFN measurement was conducted in duplicate.

### 2.9. Animal Experiments

In this study, all SPF pigs (crossbred Landrace × Duroc × Yorkshire) were purchased from a CSF-free farm in Hokkaido (Yamanaka Chikusan) that were proven to be free of antibodies against CSFV. The minimum number of animals necessary for comparison with similar previous experiments was used for each experiment: five piglets per group for the pathogenicity assessment and three piglets per group for the optimal infectious dose and vaccine efficacy studies. After piglets were introduced into the animal husbandry facility, an acclimation period of 2 days was required before they could be used for each experiment. No criteria were prepared for including or excluding animals (or experimental units), resulting in no inclusion and exclusions during the experiment.

#### 2.9.1. Pathogenicity Assessment of Two Recombinant Viruses in Pigs

Two recombinant viruses, vGPE^−^/HiBiT/Mie E2 and vGPE^−^/HiBiT/Mie E2/PAPeV E^rns^, were experimentally infected in pigs independently to assess their pathogenicity. Five 2-week-old SPF pigs per independent experiment were inoculated intramuscularly with 10^7.0^ TCID_50_ of each virus. Body temperature and clinical scores for each pig were monitored daily according to a scoring system established previously [18]. Investigator for the scoring system was blinded to the inoculated viruses. The blood samples were collected in tubes containing EDTA (Terumo Corp., Tokyo, Japan) for virus titration and blood count at 0, 3, 5, 7, 9, 11, and 14 dpi and in those containing blood coagulation factor (Terumo) for serum preparation at 0 and 14 dpi. The total number of white blood cells (WBCs) and platelets was counted using a pocH-100iV Diff apparatus (Sysmex Corp., Hyogo, Japan). Each experiment was completed at 14 dpi, and pigs were euthanized. For the virus titration, tissues from the brains, tonsils, spleens, kidneys, adrenal glands, colons, and mesenteric lymph nodes were collected aseptically. The tissue samples were homogenized in EMEM, and a 10% suspension was prepared for virus titration. The virus titers were calculated according to Reed and Muench’s [15] equation, which was TCID_50_/mL (blood) or TCID_50_/g (tissue).

#### 2.9.2. Evaluation of the Optimal Infectious Dose of vGPE^−^/HiBiT/Mie E2/PAPeV E^rns^ in Pigs

To evaluate the optimal dose of vGPE^−^/HiBiT/Mie E2/PAPeV E^rns^ in pigs for vaccination, nine 2-week-old SPF pigs were randomly divided into three subgroups (*n* = 3). Each group was inoculated with three different doses: 10^2.0^, 10^3.0^, and 10^4.0^ TCID_50_ of the virus, respectively. The clinical scores were monitored daily for 21 days, as described above. The blood samples were collected for virus titration and blood count at 0, 3, 5, 7, 9, 11, 14, and 21 dpi and for serum preparation at 0, 7, 14, and 21 dpi. All pigs were euthanized at 21 dpi, and tissues from the tonsils and kidneys were collected aseptically. The virus titers of the blood and tissue homogenates were calculated as TCID_50_/mL (blood) or TCID_50_/g (tissue). The levels of CSFV-specific neutralizing antibodies in pigs at 0, 7, 14, and 21 dpi were evaluated by SNT as previously described.

#### 2.9.3. Evaluation of the Vaccine Efficacy of vGPE^−^/HiBiT/Mie E2/PAPeV E^rns^ Against the Challenge with a Field Strain

The vaccine efficacy of vGPE^−^/HiBiT/Mie E2/PAPeV E^rns^ was evaluated by a challenge study with a field strain. Nine 2-week-old SPF pigs were randomly divided into three groups (*n* = 3): vGPE^−^/HiBiT/Mie E2/PAPeV E^rns^-vaccinated, vGPE^−^-vaccinated, and control groups. Pigs in each vaccinated group were injected intramuscularly with 10^3.0^ TCID_50_ of vGPE^−^/HiBiT/Mie E2/PAPeV E^rns^ or vGPE^−^ and with PBS. At 7 days postvaccination (dpv), all pigs were intranasally inoculated with 10^6.0^ TCID_50_ of Mie/2019 and monitored daily for body temperature and clinical scores. To monitor virus replication in the blood and the variation of blood cells, the blood samples were collected at −7, 0, 3, 5, 7, 9, 11, and 14 days post-challenge (dpc). To detect CSFV-specific neutralizing antibodies, the serum samples were collected at −7, 0, and 14 dpc. Each experiment was completed at 14 dpc, and pigs were euthanized. Tonsils, brains, spleens, kidneys, adrenal glands, colons, and mesenteric lymph nodes were collected aseptically for virus titration. The virus titer of the blood and organ samples was measured as described above.

### 2.10. Differentiation of Vaccine and Challenge Strains Recovered from Pig Samples

Viruses isolated from blood and tissue homogenates in the vaccine efficacy study were identified by reverse transcription-PCR (RT-PCR) and sequencing of the PCR product as vaccine or challenge strains. Viral RNA from the isolated viruses was extracted using the TRIzol LS reagent (Thermo Fisher Scientific). The cDNA was synthesized from extracted RNA by reverse transcription using a Random Primer (N)9 (TaKaRa Bio) and SuperScript III Reverse Transcriptase (Thermo Fisher Scientific). Subsequently, PCR was performed with KOD Fx Neo (TOYOBO, Osaka, Japan), and the primers are listed in Appendix A. The presence and size of each PCR product were confirmed by agarose gel electrophoresis.

### 2.11. A Dual Immunochromatographic (IC) Test Strip to Detect E2 and E^rns^ Antibodies Against CSFV

The swine serum samples obtained from the above experiments were used to assess the applicability of a dual IC test strip to detect E2 and E^rns^ antibodies [11] for DIVA. The serum samples were diluted with dilution buffer at a ratio of 1:1, and four drops of diluents were applied on the IC test strip. The diluent was applied to the sample drop site of a dual IC test strip. Positive or negative results were judged after 15 min incubation at RT. A positive line in the control judgment region (C) indicates that the IC test trip has been performed properly. The positive lines in the control and test judgment lines (1 and 2) indicated a positivity for antibodies against E2 and E^rns^, respectively.

### 2.12. Statistical Analysis

Microsoft Excel 365 (Microsoft Corp., Redmond, WA, USA) was used for statistical analyses. In Figure 2, the two-sided Student’s *t*-test was applied to analyze the geometric averages and variances of the data sets. In Appendix A, a one-way analysis of variance was performed. This analysis was followed by the Student’s *t*-test with Bonferroni correction.

### 2.13. Ethics Statement

The animal experiments were authorized by the Institutional Animal Care and Use Committee of the Faculty of Veterinary Medicine, Hokkaido University (approval nos. 18-0038 and 23-0029, approved on 26 March 2018 and 23 March 2023, respectively). In addition, all procedures were conducted according to the guidelines of this committee. All experiments were conducted under the guidance of this committee. The Faculty of Veterinary Medicine at Hokkaido University has maintained accreditation of the Association for Assessment and Accreditation of Laboratory Animal Care International since 2007.

All experiments were authorized by the Safety Committee on Genetic Recombination Experiments of Hokkaido University and the Ministry of Education, Culture, Sports, Science, and Technology (approval nos. 2020-009 and 6-Monkashin-Dai-460, approved on 8 June 2021, and 19 August 2024, respectively) for genetically modified mutant viruses generated by reverse genetics.

## 3. Results

### 3.1. The Antigenic Gap Between the Vaccine and Field Strains

To investigate the neutralizing activity of vaccine-induced antibodies to the current field isolate, the serum samples were collected from 20 pigs vaccinated with GPE^−^ and subjected to SNT against vGPE^−^ and Mie/2019, respectively (Figure 2). Neutralizing antibody titers of each pig serum with the Mie/2019 strain were lower than those with the vGPE^−^ strain. The geometric means of ND_50_ against vGPE^−^ and Mie/2019 were 2^8.0^ (256) and 2^5.4^ (42.0), respectively. This result implied a significant antigenic gap between GPE^−^ and a current filed strain in Japan.

### 3.2. Generation of Recombinant Viruses Antigenically Close to a Field Strain

It was expected that the antigenic match between the vaccine and field strains would confer effective protection against CSF in pigs. In CSFV infections, neutralizing antibodies are produced mainly against the viral glycoprotein E2. To match the antigenicity between the vaccine and field strains, an infectious cDNA clone, pGPE^−^/HiBiT/Mie E2, was constructed by substituting the E2-encoding region of pGPE^−^/HiBiT with that of Mie/2019. Moreover, to introduce a marker property, the E^rns^-encoding region of pGPE^−^/HiBiT/Mie E2 was replaced with that of PAPeV, namely pGPE^−^/HiBiT/Mie E2/PAPeV E^rns^. The full-length viral RNA was synthesized from each cDNA clone and introduced to SK-L cells through electroporation. After 3 days of incubation, the focal expression of the viral NS3 protein was confirmed by IPX (Appendix A), demonstrating the successful virus rescue of vGPE^−^/HiBiT/Mie E2 and vGPE^−^/HiBiT/Mie E2/PAPeV E^rns^.

### 3.3. Antigenicity of the Rescued Viruses

To assess the antigenicity of the generated recombinant viruses, the antiserum against vGPE^−^/HiBiT/Mie E2 was prepared. Originally, antiserum against the field strain should have been prepared; however, antiserum against this recombinant virus was prepared for the antigenic analysis because the preparation of antiserum against a field virulent strain is difficult due to the very low antibody response caused by immunosuppression. The SNT using anti-GPE^−^ serum showed that the ND_50_ against vGPE^−^/HiBiT/Mie E2, vGPE^−^/HiBiT/Mie E2/PAPeV E^rns^, and the field strain Mie/2019 were 128.0, 38.1, and 76.1, respectively. These titers were 4.8-, 16-, and 8-fold lower than those against homologous viruses, vGPE^−^/HiBiT (ND_50_ = 608.9), respectively (Table 1). In contrast, the titers of anti-vGPE^−^/HiBiT/Mie E2 serum against E2-swapped viruses and Mie/2019 were 152.2, 64.0, and 76.1, respectively. These titers were 22.7-, 9.6-, and 11.4-fold higher than those against the vaccine-derived virus, vGPE^−^/HiBiT (ND_50_ = 6.7). These results indicate that swapping the E2 sequence contributed to the generation of recombinant viruses, which were antigenically similar to the field strain Mie/2019.

### 3.4. In Vitro Characterization of the Rescued Viruses

To evaluate the growth and genetic stability of the rescued viruses, vGPE^−^/HiBiT/Mie E2 and vGPE^−^/HiBiT/Mie E2/PAPeV E^rns^ were independently passaged in SK-L cells five times. Both viruses stably exhibited comparable replication abilities to vGPE^−^ [10] even after serial passages in cells (Appendix A). To assess the genetic instability of vGPE^−^/HiBiT/Mie E2/PAPeV E^rns^ of the HiBiT tag-inserted site, virus titration was conducted by the luciferase assay and IPX. No significant differences were observed between the titers measured using the two methods in both recombinant viruses (Appendix A).

Regarding genetic stability, no nucleotide substitution was confirmed in vGPE^−^/HiBiT/Mie E2 before and after five passages. In addition, vGPE^−^/HiBiT/Mie E2/PAPeV E^rns^ exhibited no nucleotide substitution in the E2-encoding region after five passages. In contrast, one nucleotide substitution, alanine (A) to guanine (G), was identified in the E^rns^ region of vGPE^−^/HiBiT/Mie E2/PAPeV E^rns^ after the first passage (P1 virus), leading to an amino acid change from glutamic acid to lysine at position 92. Among the P2 to P5 viruses, various nucleotides were confirmed at this position; both A and G signals overlapped in the Sanger sequence. In addition, the sequence encoding the HiBiT tag could not be determined due to the overlapping of multiple base signals at this site in the Sanger sequencing in the P2 to P5 viruses, suggesting the genetic instability of vGPE^−^/HiBiT/Mie E2/PAPeV E^rns^ in this region. These results indicated that vGPE^−^/HiBiT/Mie E2/PAPeV E^rns^ should be genetically unstable and should not be further used after several passages. Therefore, the P1 of vGPE^−^/HiBiT/Mie E2/PAPeV E^rns^ was used in the following assays.

To assess the impact of substituting glycoprotein E2 on viral growth kinetics and IFN-α/β production in swine cells, two recombinant viruses and vGPE^−^/HiBiT were inoculated into SK-L cells at an MOI of 0.001. The infectivity titers of vGPE^−^/HiBiT/Mie E2 and vGPE^−^/HiBiT/Mie E2/PAPeV E^rns^ were significantly higher than that of vGPE^−^/HiBiT (Appendix A). Correlating with the significantly higher growth of the recombinant viruses in SK-L cells compared to vGPE^−^/HiBiT, IFN-α/β production by vGPE^−^/HiBiT/Mie E2 and vGPE^−^/HiBiT/Mie E2/PAPeV E^rns^ was significantly higher than that of vGPE^−^/HiBiT at 6 and 7 dpi (Appendix A).

### 3.5. Pathogenicity of Generated Recombinant Viruses in Piglets

To assess the pathogenicity of vGPE^−^/HiBiT/Mie E2 and vGPE^−^/HiBiT/Mie E2/PAPeV E^rns^ in pigs, 10^7.0^ TCID_50_ of each virus was intramuscularly inoculated in five 2-week-old piglets, and their body temperatures, clinical signs, and blood cell counts were monitored for 14 days. Almost all piglets inoculated with vGPE^−^/HiBiT/Mie E2 or vGPE^−^/HiBiT/Mie E2/PAPeV E^rns^ did not exhibit fever throughout the experimental period, except for one piglet inoculated with vGPE^−^/HiBiT/Mie E2 that developed a fever of 40.5 °C at 9 dpi (Appendix A). The clinical signs specific to CSF were rarely observed in piglets inoculated with each virus. The maximum clinical score was 1, including slightly reddened eyelids and slow eating (Appendix A). One piglet (#399) in the vGPE^−^/HiBiT/Mie E2-inoculated group showed a reduction of WBCs and platelets. In contrast, neither leukopenia (≤8 × 10^3^/µL, normal range in adult: 12–30 × 10^3^/µL) nor thrombocytopenia (≤10 × 10^4^/µL, normal range in adult: 25–60 × 10^4^/µL) was observed in any piglet inoculated with vGPE^−^/HiBiT/Mie E2/PAPeV E^rns^ (Appendix A). In this study, 2-week-old piglets after weaning were used in each experiment. It is known that the white blood cell and platelet counts are relatively higher in young animals than in adult ones, and these numbers measured at 0 dpi were judged to be normal.

The blood and tissue samples were collected from the piglets to evaluate viral growth in piglets (Table 2). In the vGPE^−^/HiBiT/Mie E2-inoculated group, all piglets developed transient viremia between 3 and 9 dpi, and the maximum virus titer was 10^2.9^ TCID_50_/mL. In organ samples, the virus was recovered from the tonsils of all piglets with titers from 10^3.0^ to 10^4.6^ TCID_50_/g. The virus was also recovered from the kidneys (*n* = 1), adrenal glands (*n* = 1), and mesenteric lymph nodes (*n* = 1). In the vGPE^−^/HiBiT/Mie E2/PAPeV E^rns^ group, transient viremia between 3 and 9 dpi was observed in all piglets with 10^2.3^ TCID_50_/mL as the maximum virus titer. The virus titers recovered from the tonsils were from ≤10^1.8^ to 10^4.5^ TCID_50_/g. No virus was detected from the other organs, except for the spleen of one piglet. In total, two viruses, vGPE^−^/HiBiT/Mie E2/and vGPE/HiBiT/Mie E2/PAPeV, replicated equivalently in piglets. Compared to a previous study using the parental strain, vGPE^−^ [19], vGPE^−^/HiBiT/Mie E2, and vGPE^−^/HiBiT/Mie E2/PAPeV E^rns^ showed higher growth potential in piglets. Because vGPE^−^/HiBiT/Mie E2/PAPeV E^rns^ possesses a potential immunological marker, vGPE^−^/HiBiT/Mie E2/PAPeV E^rns^ was used for further in vivo assessment as a vaccine candidate.

### 3.6. Optimal Dose of vGPE^−^/HiBiT/Mie E2/PAPeV E^rns^ for Vaccination

The current commercial vaccine strain, GPE^−^, is subject to injection with 10^3.0^ TCID_50_/pig in the field. To determine the optimal dose of vGPE^−^/HiBiT/Mie E2/PAPeV E^rns^ for vaccination, each group of three piglets was intramuscularly inoculated with 10^2.0^, 10^3.0^, or 10^4.0^ TCID_50_ of vGPE^−^/HiBiT/Mie E2/PAPeV E^rns^, and viral growth and vaccine-induced antibody response in piglets were evaluated for 21 days. During the experiment, none of the piglets exhibited clinical signs specific to CSFV infection, including leukopenia and thrombocytopenia (Appendix A). The reason for the post-vaccination increase in white blood cell count was assessed as an immune response to vaccination and not a serious adverse reaction. To monitor viremia, the blood samples were collected after virus inoculation (Table 3). The seven piglets of the three different doses were in the state of transient viremia between 5 and 11 dpi. No virus was detected in the blood collected from the other two piglets, which were inoculated with the lowest dose. However, the virus was recovered from the tonsils collected at 21 dpi, even from these two. Taken together, all nine piglets inoculated with three different doses were infected with the recombinant virus (Table 3).

To evaluate the humoral immune response, SNT was performed using serum samples collected at 0, 7, 14, and 21 dpi. The neutralizing antibodies were detected at 14 dpi in three piglets inoculated with 10^3.0^ TCID_50_ of vGPE^−^/HiBiT/Mie E2/PAPeV E^rns^. At 21 dpi, neutralizing antibodies were induced in all piglets, except for one piglet inoculated with 10^2.0^ TCID_50_, and the ND_50_ values were 2.8–32.0 (Table 3). Results indicated that vGPE^−^/HiBiT/Mie E2/PAPeV E^rns^ replicated well and induced the humoral immune response in almost all piglets. From the above results, 10^3.0^ TCID_50_/pig was employed as the optimal vaccination dose in further studies, the same vaccine dose as the current vaccine strain, GPE^−^.

### 3.7. Vaccine Efficacy of vGPE^−^/HiBiT/Mie E2/PAPeV E^rns^

Nine 2-week-old SPF pigs were divided into three groups (*n* = 3): vGPE^−^/HiBiT/Mie E2/PAPeV E^rns^-vaccinated, vGPE^−^-vaccinated, and control groups. All pigs were intranasally inoculated with 10^6.0^ TCID_50_ of Mie/2019 on 7 dpv. The piglets vaccinated with vGPE^−^/HiBiT/Mie E2/PAPeV E^rns^ or vGPE^−^ had neither symptoms nor fever after the intranasal challenge with Mie/2019 (Appendix A). In the control group, two of the three piglets showed slightly less liveness between 9 and 13 dpc and had a fever at 14 dpc, although no other clear clinical signs were observed (Appendix A). Neither leukopenia nor thrombocytopenia was confirmed in either vaccinated group, whereas a decrease in WBCs and platelets after the challenge was observed in the control group (Appendix A).

Although clinical symptoms of CSF were rarely observed in nonvaccinated piglets, viremia was confirmed from 3 dpc until the end of the experiment with the increase of virus titer (Table 4). In addition, viruses with high titers were detected in all collected organs. In the vGPE^−^/HiBiT/Mie E2/PAPeV E^rns^-vaccinated group, viruses were recovered from the blood between 0 and 3 dpc. However, these recovered viruses were vGPE^−^/HiBiT/Mie E2/PAPeV E^rns^, suggesting that transient viremia by vaccination remained even after the virus challenge. Compared to the control group, virus recovery was limited in the blood and organ samples of all vaccinated piglets (Table 4). Taken together, vGPE^−^/HiBiT/Mie E2/PAPeV E^rns^ showed sufficient immunity to vaccine efficacy against a field strain comparable to that induced by vGPE^−^ vaccine.

### 3.8. Antibody Detection and Evaluation of DIVA Properties in Animal Experiments

The efficacy of vGPE^−^/HiBiT/Mie E2/PAPeV E^rns^ against the challenge of field strain Mie/2019 in piglets was equivalent to that in piglets vaccinated with vGPE^−^. In this study, neutralizing antibody titers were also monitored for each group (Table 5). The antibody titers in the vaccinated group on the day of the challenge, i.e., 7 dpv, were below the detection limit. On 14 dpc, the neutralizing antibody was detected in the group vaccinated with vGPE^−^/HiBiT/Mie E2/PAPeV E^rns^, which was almost the same for the vaccine and field strains. This is reasonable, considering that the antigenicity of vGPE^−^/HiBiT/Mie E2/PAPeV E^rns^ is closely related to that of the challenge strain Mie/2019. In contrast, in the vGPE^−^-vaccinated group, because of the lower homology of the amino acid sequence in the E2 protein between the vaccine and challenge strains, antibody titers against the vaccine strain were sufficiently high but were lower against the field strain. No specific antibody was detected from nonvaccinated piglets (Table 5), although it was confirmed that the challenge virus replicated well in these piglets (Table 4).

An IC test strip developed in our previous work [11] was used to detect antibodies against CSFV E^rns^ and E2 proteins in serum to confirm the applicability of the marker function that identified antibodies against vGPE^−^/HiBiT/Mie E2/PAPeV E^rns^ (Table 6). As a group, antibodies against CSFV E2, but not E^rns^, were detected in the vaccination group with vGPE^−^/HiBiT/Mie E2/PAPeV E^rns^ (pig IDs #418 and #419). Meanwhile, in the challenge experiment with the field virus, antibodies against CSFV E^rns^ were also detected at 14 dpc in groups vaccinated with vGPE^−^/HiBiT/Mie E2/PAPeV E^rns^ (#425) and vGPE^−^ (#426–428), suggesting that the multiplication of the challenge virus in vaccinated pigs was serologically confirmed. The antibody-positive result against CSFV E^rns^ was also obtained in the group challenged with the field strain without vaccination (#431). In addition, no CSFV E2 antibody was detected in this control (PBS) group because of the lower immunological response to the infection with a virulent field strain. Taken together, these results suggest that vGPE^−^/HiBiT/Mie E2/PAPeV E^rns^ has the potential to fulfill the DIVA function despite limitations in the number of pigs used and the duration of the animal experiment.

## 4. Discussion

To control CSF, the prophylactic vaccination measure has been applied in pig farms across Japan since 2019 [20]. However, 35 CSF cases had been reported in vaccinated farms as of the end of November 2024. The antigenic gap between the vaccine strain, GPE^−^, and current field strains, indicated in this study, was considered as one of the reasons for these outbreaks by reducing the vaccine efficacy. Thus, this study aimed to develop new CSF vaccine candidates based on vGPE^−^ that antigenically match current field strains. Because glycoprotein E2 is an immunogenically dominant protein, eliciting neutralizing antibodies in pestiviruses [21,22], vGPE^−^/HiBiT/Mie E2 was generated by replacing the E2 region of vGPE^−^ with that derived from a field strain, Mie/2019, to raise the immunity by vaccination. Furthermore, considering the application of this recombinant virus as a DIVA vaccine, vGPE^−^/HiBiT/Mie E2/PAPeV E^rns^ was generated from vGPE^−^/HiBiT/Mie E2 by the additional substitution of the glycoprotein E^rns^ with that of PAPeV as an immunological marker. Importantly, the generated recombinant viruses successfully maintained the antigenicity of the original Mie/2019. Both recombinant viruses showed higher replication abilities in piglets than vGPE^−^. Because vGPE^−^/HiBiT/Mie E2/PAPeV E^rns^ possesses a potential immunological marker, which is useful for the implementation of the DIVA vaccine, the P1 of vGPE^−^/HiBiT/Mie E2/PAPeV E^rns^ was used for further in vivo study. Through the challenge study, vGPE^−^/HiBiT/Mie E2/PAPeV E^rns^ conferred sufficient immunity to protect against the challenge with a recent CSFV isolate, comparable efficacy as vGPE^−^. In previous studies of orthoflaviviruses belonging to the same *Flaviviridae* family, several chimeric viruses harboring the E protein derived from target strains, which is the major antigenicity determinant in orthoflaviviruses, were investigated for potentials as live-attenuated vaccines [23,24]. Accordingly, installing the E2 protein of the prevalent field strains into the vaccine strain is a promising strategy for establishing CSF vaccine strains with higher potency, which is a likely applicable concept for vaccine development against other pestivirus infections.

The homology of E2 between Mie/2019 and vGPE^−^ is 89.3% in amino acids; there are 40 amino acid differences, and 23 of them are located in the antigenic region, suggesting an antigenic gap between the vaccine and field strains. A previous study suggested that the amino acid substitutions D705N, L709P, G713E, N723S, and S779A were mainly responsible for the antigenic gap between the C-strain (genotype 1 vaccine strain) and subgenotype 2.1 strains [6]. Among these amino acid positions, the substitutions L709P, G713E, and N723S were also identified in Mie/2019 compared with vGPE^−^, which probably contributes as well to the different antigenicities recognized between them. Further studies to clarify the contributions of other amino acid substitutions to the antigenicity would provide significant insights into the antigenic structure of the CSFV E2 glycoprotein. It has been 6 years since the reintroduction of CSFV in Japan in 2018. During this period, CSF cases had been sporadically confirmed, especially from wild boars [7,8], suggesting the continuous evolution of viruses. A previous epidemiological study suggested that genetic diversity among field CSFVs in Japan depends on the area and time of virus isolation [25]. Even in the same subgenotypes, antigenic variations were identified [6]. Therefore, continuous monitoring of the antigenicity of prevalent CSFVs is necessary to update CSF vaccine strains in the future.

Because the safety of the host species is the minimum requirement for live-attenuated vaccine strains, the pathogenicity of newly generated viruses in pigs should be assessed. This study indicated that the substitution of the E2-encoding region derived from a field strain resulted in higher viral replication in piglets than vGPE^−^ [19]. The E2 glycoprotein is involved in the viral attachment and entry into host cells [26], and several studies have identified critical amino acids in the E2 for CSFV virulence (Appendix A). The T830A substitution of vGPE^−^ enhanced viral spreading and replication in swine-derived cells and showed higher pathogenicity in pigs [14]. T745I with a combination of M979K substitution increased the cell-to-cell spreading ability of the C-strain in vitro [27]. The highly virulent strain Shimen was attenuated in vivo by deglycosylation at position 986 with the A988T substation [28]. The E2 of Mie/2019 possesses virulent-type amino acids at these positions (Appendix A). Thus, these amino acid residues will be potential targets for additional genetic manipulations to attenuate the generated vaccine candidates and ensure their safety.

Most pestiviruses are non-cytopathogenic, resulting in no morphological changes in cultured cells during virus infection. The installation of the HiBiT tag into pestiviruses enabled more rapid and easier determination of virus propagation in cells compared to the conventional method such as IPX. In this study, vGPE^−^/HiBiT/Mie E2 was genetically stable after serial passages in cells, consistent with the previous report that neither mutation nor deletion at the HiBiT tag-inserted site was found in vGPE^−^/HiBiT [13]. In contrast, the HiBiT tag inserted at the N-terminus of PAPeV E^rns^ was genetically unstable. In addition, one amino acid substitution was found in the PAPeV E^rns^. This genetic instability was inferred due to the combination of the HiBiT tag in the PAPeV E^rns^, which was supported by the result that no nucleotide mutation was confirmed in vGPE^−^/HiBiT/Mie E2 and vGPE^−^/PAPeV E^rns^ [10] after five passages. To further exploit the advantages of the HiBiT system, subsequent work will be required to determine the optimal insertion positions of the HiBiT tag to avoid the interference of the original viral and inserted proteins. Alternatively, a new candidate strain without the HiBiT tag, namely vGPE^−^/Mie E2/PAPeV E^rns^, may be promising as a genetically stable vaccine strain. Further analysis is essential to assess these points.

In our experimental setting for evaluating vaccine efficacy, piglets were challenged at 7 dpv before the neutralizing antibodies were detected (Table 5). In this study, neutralizing antibody titers were detected at 14 dpc from all pigs vaccinated with vGPE^−^/HiBiT/Mie E2/PAPeV E^rns^ and vGPE^−^.

This rapid immune response in neutralizing antibodies in vaccinated pigs could contribute to clinical and virological protections, as discussed previously [10]. Notably, neutralizing antibodies at 14 dpc against the Mie/2019 strain were higher in the piglets immunized with vGPE^−^/HiBiT/Mie E2/PAPeV, suggesting that the immunization induced by the vGPE^−^/HiBiT/Mie E2/PAPeV produced the antibody that efficiently neutralizes the current field strain. However, from the clinical and virological aspects, we could not show that vGPE^−^/HiBiT/Mie E2/PAPeV E^rns^ was more effective than conventional vGPE^−^. Generally, CSFV strains are categorized into the following three major genotypes: cross-reactive neutralizing antibodies inhibit the virus growth of the genotype 2 strain in the pigs vaccinated properly with genotype 1 [29]. In addition, the limited replication of the challenge virus may be achieved partially by cellular-mediated immunity elicited by the live-attenuated vaccine [10,30]. Furthermore, given the existence of vGPE^−^/HiBiT/Mie E2/PAPeV E^rns^ and vGPE^−^ even after the challenge, the replicating vaccine strains could partially inhibit the growth of the challenge virus by the phenomenon of superinfection exclusion [31]. To prove the hypothesis that the novel vaccine candidates in this study are more effective against the genotype 2 field strains, it is necessary to design the experiment in an environment where cellular immunity and the superinfection exclusion phenomenon are not induced by vaccine injection, and only weak humoral immunity works to inhibit the growth of the challenge virus. To achieve this, further study to challenge nonvaccinated piglets with maternal antibodies derived from sows vaccinated with conventional GPE^−^ or a new vaccine candidate is required, as demonstrated in the previous report [32]. However, we designed the current experimental design as a preliminary stage before experiments using piglets with only maternal antibodies since conducting this experiment requires a substantial breeding system, long lead time, and human resources.

In addition, to evaluate the validity of the results in animal experiments, the number of pigs in each group should be increased, and the immune response of each group and its protective effect against a field CSFV strain should be statistically evaluated in more detail. By using samples taken in these additional experiments, more serum samples should be used to evaluate the properties of DIVA.

In conclusion, this study demonstrated that replacing the E2 encoding sequence achieved antigenic matching between the vaccine and target field strains. The potency of E2-swapped vGPE^−^ as a vaccine candidate and its DIVA properties were evaluated in animal experiments. For practical use, further studies are needed to enhance the utility of vaccine candidates. However, this study may provide a new strategy for improving CSF vaccines, which will likely contribute to the eradication of CSF in Japan.

## Figures and Tables

**Figure 1 vaccines-13-00188-f001:**
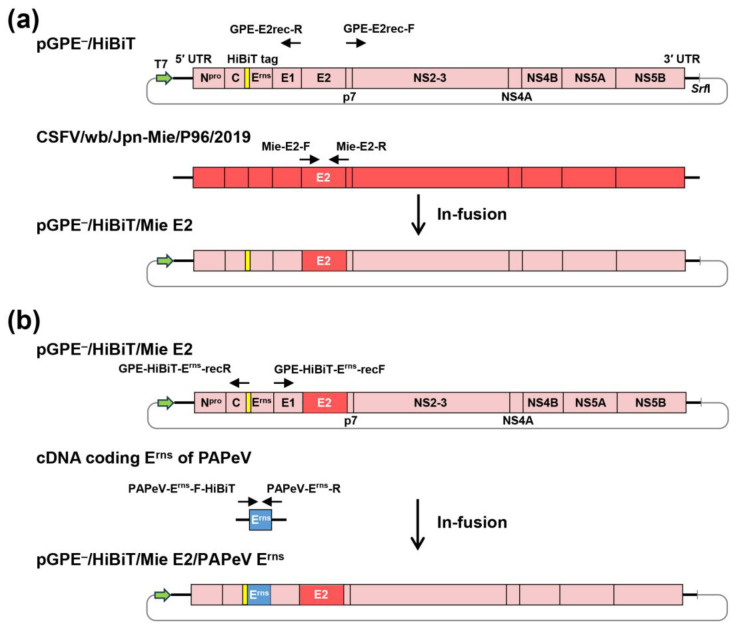
Construction of the recombinant plasmids. The construction procedure of (**a**) pGPE^−^/HiBiT/Mie E2 and (**b**) pGPE^−^/HiBiT/Mie E2/PAPeV E^rns^ are shown. The coding regions of the recombinant viruses are displayed as boxes divided by each protein. (**a**) E2 of Mie/2019 and pGPE^−^/HiBiT other than the E2-encoding region, (**b**) E^rns^ of PAPeV and pGPE^−^/HiBiT/Mie E2 other than E^rns^-encoding region were amplified by the indicated primers in Appendix A, respectively. Each set of polymerase chain reaction (PCR) products was fused with the overlapping regions. Yellow boxes indicate the HiBiT tag. T7, T7 promoter; *Srf*I, *Srf*I restriction enzyme site.

**Figure 2 vaccines-13-00188-f002:**
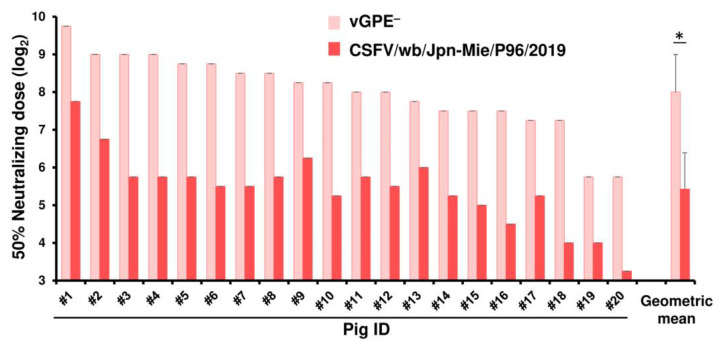
Antibody titers of pigs vaccinated with GPE^−^ against the vaccine and field strains. SNT was performed using the serum samples from 20 pigs vaccinated with GPE. Each antibody titer against vGPE^−^ is indicated by pink bars, and that against Mie/2019 is shown by red bars. Each serum sample was tested with four wells per virus to calculate their antibody titers. The geometric mean value of titers of the 20 serum samples is indicated, with error bars representing standard deviations (*n* = 20). * *p* < 0.05 between vGPE^−^ and Mie/2019.

**Table 1 vaccines-13-00188-t001:** Neutralizing antibody titers of pig anti-CSFV sera against different CSFV strains.

Viruses	ND_50_ of Antisera	
	Anti-GPE^−^	Anti-vGPE^−^/HiBiT/Mie E2
vGPE^−^/HiBiT	608.9 ^1^	6.7
vGPE^−^/HiBiT/Mie E2	128.0	152.2
vGPE^−^/HiBiT/Mie E2/PAPeV E^rns^	38.1	64.0
Mie/2019	76.1	76.1

^1^ Each serum sample was tested with four wells per virus to calculate ND_50_. Underlines indicate homologous combinations between serum and indicator virus.

**Table 2 vaccines-13-00188-t002:** Virus recovery from pigs infected with vGPE^−^/HiBiT/Mie E2 or vGPE^−^/HiBiT/Mie E2/PAPeV E^rns^.

Virus	Pig ID	Blood (log_10_ TCID_50_/mL) ^1^ at dpi						Tissue (log_10_ TCID_50_/g) ^2^						
		3	5	7	9	11	14	To.	Br.	Sp.	Kid.	Ad.	Co.	Ly.
vGPE^−^/HiBiT/Mie E2	#398	−	+	+	−	−	−	3.8	−	−	−	−	−	−
	#399	−	+	+	−	−	−	3.5	−	−	−	−	−	−
	#400	−	1.0	2.1	+	−	−	3.0	−	−	+	−	−	−
	#401	−	2.9	2.8	1.0	−	−	3.8	−	−	−	+	−	−
	#402	1.0	1.8	1.8	+	−	−	4.6	−	−	−	−	−	+
vGPE^−^/HiBiT/Mie E2/PAPeV E^rns^	#409	1.0	1.8	1.8	−	−	−	4.5	−	−	−	−	−	−
	#410	−	+	+	+	−	−	+	−	−	−	−	−	−
	#411	−	+	1.8	+	−	−	4.0	−	+	−	−	−	−
	#412	+	+	+	−	−	−	2.8	−	−	−	−	−	−
	#413	+	2.3	≤1.1	1.0	−	−	2.6	−	−	−	−	−	−

^1^ −, not isolated; +, isolated in a 6-well plate and lower than the detection limit of TCID_50_ (10^0.8^ TCID_50_/mL) in a 96-well plate. ^2^ −, not isolated; +, isolated in a 6-well plate and lower than the detection limit of TCID_50_ (10^1.8^ TCID_50_/g) in a 96-well plate. To., tonsil; Br., brain; Sp., spleen; Kid., kidney; Ad., adrenal gland; Co., colon; and Ly., mesenteric lymph node.

**Table 3 vaccines-13-00188-t003:** Virus recovery from pigs inoculated with different doses of vGPE^−^/HiBiT/Mie E2/PAPeV E^rns^ and their neutralizing antibody titers.

Virus Titer (TCID_50_)	Pig ID	Blood (log_10_ TCID_50_/mL) ^1^ at dpi							Tissue (log_10_TCID_50_/g) ^2^		ND_50_ ^3^ at dpi			
		3	5	7	9	11	14	21	To.	Kid.	0	7	14	21
10^2.0^	#414	−	−	−	−	−	−	−	1.0	−	<1.0	<1.0	<1.0	8.0
	#415	−	−	+	≤1.1	+	−	−	+	−	<1.0	<1.0	<1.0	2.8
	#416	−	−	−	−	−	−	−	+	−	<1.0	<1.0	<1.0	<1.0
10^3.0^	#417	−	−	−	−	1.0	−	−	−	−	<1.0	<1.0	1.0	8.0
	#418	−	+	+	+	−	−	−	+	−	<1.0	<1.0	1.0	22.6
	#419	−	+	+	+	−	−	−	−	−	<1.0	<1.0	5.6	32.0
10^4.0^	#420	−	−	+	1.0	+	−	−	+	−	<1.0	<1.0	<1.0	8.0
	#421	−	−	1.0	−	−	−	−	−	−	<1.0	<1.0	<1.0	32.0
	#422	−	−	≤1.1	−	−	−	−	+	−	<1.0	<1.0	<1.0	22.5

^1^ −, not isolated; +, isolated in a 6-well plate and lower than the detection limit of TCID_50_ (10^0.8^ TCID_50_/mL) in a 96-well plate. ^2^ −, not isolated; +, isolated in a 6-well plate and lower than the detection limit of TCID_50_ (10^1.8^ TCID_50_/g) in a 96-well plate. ^3^ ND_50_ was determined using two duplicate wells for each serum sample.

**Table 4 vaccines-13-00188-t004:** Virus recovery from blood and organ samples of piglets in the challenge study.

Vaccine Strain	Pig ID	Blood (log_10_ TCID_50_/mL) ^1^ at dpc								Tissue (log_10_ TCID_50_/g) ^2^						
		−7	0	3	5	7	9	11	14	To.	Br.	Sp.	Kid.	Ad.	Co.	Ly.
vGPE^−^/HiBiT/Mie E2/PAPeV E^rns^	#423	−	+ ^3^	−	−	+ ^3^	−	−	−	+ ^3^	−	−	−	−	−	−
	#424	−	+ ^3^	−	−	−	−	−	−	+ ^3^	−	−	−	−	−	−
	#425	−	−	+ ^3^	−	−	−	−	−	2.0 ^6^	−	−	−	−	−	−
vGPE^−^	#426	−	−	−	−	−	−	−	−	2.0 ^6^	−	−	−	−	−	+ ^5^
	#427	−	−	−	−	−	−	−	−	+ ^4^	−	−	−	−	−	−
	#428	−	−	−	−	−	−	+ ^5^	−	+ ^5^	−	−	−	−	−	−
PBS	#429	−	−	+	2.6	3.3	4.3	5.6	6.3	5.9	4.0	6.0	5.3	5.3	5.3	6.8
	#430	−	−	+	2.1	2.3	3.8	5.3	5.8	6.8	3.3	6.0	4.6	4.5	5.8	5.9
	#431	−	−	2.1	2.8	3.0	4.6	5.0	5.8	7.0	3.2	6.5	5.0	5.0	6.3	7.8

^1^ −, not isolated; +, isolated in a 6-well plate and lower than the detection limit of TCID_50_ (10^0.8^ TCID_50_/mL) in a 96-well plate. ^2^ −, not isolated; +, isolated in a 6-well plate and lower than the detection limit of TCID_50_ (10^1.8^ TCID_50_/g) in a 96-well plate. ^3–6^ The isolated viruses were confirmed as vGPE^−^/HiBiT/Mie E2/PAPeV E^rns^ (3), vGPE^−^ (4), Mie/2019 (5), or both of the vaccine strain and Mie/2019 (6) by RT-PCR using primers that can specifically detect each virus.

**Table 5 vaccines-13-00188-t005:** Detection of neutralizing antibodies in piglets in the challenge study.

Vaccine Strain	Pig ID	ND_50_ ^1^ Against Each Test Virus at			
		−7 dpc	0 dpc	14 dpc	
		vGPE^−^/HiBiT	Vaccine Strain ^2^	Vaccine Strain	Mie/2019
vGPE^−^/HiBiT/Mie E2/PAPeV E^rns^	#423	<1	<1	31.9	45.2
	#424	<1	<1	16.0	11.3
	#425	<1	<1	8.0	11.3
vGPE^−^	#426	<1	<1	11.3	<1.0
	#427	<1	<1	5.6	1.0
	#428	<1	<1	22.6	<1.0
PBS	#429	<1	NT	NT	<1.0
	#430	<1	NT	NT	<1.0
	#431	<1	NT	NT	<1.0

^1^ ND_50_ was determined using duplicate wells for each serum sample. ^2^ The neutralizing antibody titer against the virus used for vaccination was measured. NT, not tested.

**Table 6 vaccines-13-00188-t006:** Detection of E^rns^ and E2 antibodies using an IC test strip from representative serum samples.

Vaccinated Virus	Challenge Strain	Pig ID	Date of the Test	IC Test Strip			The SNT Titer Against ^1^	
				E^rns^	E2		Vaccine Strain	Mie/2019
vGPE^−^/HiBiT/Mie E2/PAPeV E^rns^		#417		−	−		8.0	NT
	No challenge	#418	21 dpi	−	+		22.6	NT
		#419		−	+		32.0	NT
vGPE^−^/HiBiT/Mie E2/PAPeV E^rns^		#423		−	+		31.9	45.2
	Mie/2019	#424	14 dpc	−	+		16.0	11.3
		#425		+	+		8.0	11.3
vGPE^−^		#426		+	+		11.3	<1.0
	Mie/2019	#427	14 dpc	+	−		5.6	1.0
		#428		+	−		22.6	<1.0
PBS		#429		−	−		NT	<1.0
	Mie/2019	#430	14 dpc	−	−		NT	<1.0
		#431		+	−		NT	<1.0

^1^ The SNT titers are originally given in Table 3 and Table 5. N/A, not available; +, positive; −, negative. NT, not tested.

## Data Availability

All data are contained within the article.

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
