# Peer review of "Generation of Vaccine Candidate Strains That Antigenically Match Classical Swine Fever Virus Field Strains"

_vaccines, 2025, doi:10.3390/vaccines13020188_

Round 1
Reviewer 1 Report
Comments and Suggestions for Authors
Classical swine fever (CSF) virus is the causative agent of an economically important, highly contagious disease of pigs. Reverse genetics methods make it possible to obtain modified viruses, including vaccine viruses, without the use of animals or long-term multiple passaging in cell cultures. Currently, reverse genetics plays a crucial role in formulating prevention and control strategies for CSF. Although obtaining a recombinant CSFV vaccine strain by reverse genetics is not new and has often been used previously to obtain marked vaccines against CSF, the proposed approach is interesting and relevant. The authors have experience with the CSFV for a long time. The methodology used in the articles is adequate. The article is well written and can be published. The main drawback of the manuscript is the small groups of animals. However, the authors note that further research is needed.
Author Response
Response to Reviewer 1:
Thank you for your careful review of our manuscript. The modifications made according to your comments are highlighted in purple.
- Classical swine fever (CSF) virus is the causative agent of an economically important, highly contagious disease of pigs. Reverse genetics methods make it possible to obtain modified viruses, including vaccine viruses, without the use of animals or long-term multiple passaging in cell cultures. Currently, reverse genetics plays a crucial role in formulating prevention and control strategies for CSF. Although obtaining a recombinant CSFV vaccine strain by reverse genetics is not new and has often been used previously to obtain marked vaccines against CSF, the proposed approach is interesting and relevant. The authors have experience with the CSFV for a long time. The methodology used in the articles is adequate. The article is well written and can be published. The main drawback of the manuscript is the small groups of animals. However, the authors note that further research is needed.
[Response]
We deeply appreciate your constructive comments on our paper. We have shown that recombinants of the E2 gene region can be used as antigenically matched vaccine strains against field CSFV. As the reviewers mentioned, this study has several limitations that require further study. The small number of pigs used in the experiments, as highlighted by the reviewer, represents a challenge that should be overcome in the future. We have added this point in the text.
[Revision in the text]
The text was modified in Lines 610-613.
Reviewer 2 Report
Comments and Suggestions for Authors
The submitted manuscript describes the development and evaluation of a new vaccine candidate against swine fever virus by substitution of the E2 antigen in the currently used GPE- vaccine with the E2 antigen from the wild strain.
The study is well done and clearly written. The paper presents evidence of antigenic differences between E2, the major protective antigen of swine fever virus, in the vaccine and the wild strain Mie2019, and the replacement of the antigen with a more relevant one. However, such replacement did not improve the vaccine's efficacy against field strains. Therefore, other pathogenicity factors should be considered and investigated to increase the protective effect. This should be highlighted in the paper.
Some minor points are following.
L 21 Please decipher “GPE− vaccine”, so that it is clear, for example, “based on CSFV strain deficient in glycoprotein E”. Avoid complex abbreviations which may be unknown for readers and are difficult to differentiate.
L28 The reason of PAPeV Erns substitution is not evident. Please, clarify. How Mie protein substitution with proteins from the other viruses provide protectivity against CSFV/Mie?
L132 Specify how long after vaccination of 20 pigs the sera were collected.
L300-306. “The Antigenic Gap” is expressed in Fig 2 quantitatively. This way the authors did not confirm antigen differences between the vaccine and field strains. The obtained results may be due to different expression/presentation and therefore different binding/neutralization of the same antigen in these strains. Provide explanation comments or direct confirmation, differences between their E2 antigens.
L330 Fix a typo ‘filed strain’
L333-335. I do not agree with the authors' conclusion from Table 1 results that the recombinant viruses are antigenically similar to the field strain Mie/2019. The results demonstrate that the neutralizing activity of the anti-vaccine virus sera is equal to that of the anti-recombinant virus sera (76=76). Thus, the E2 substitution is ineffective. This is also confirmed (L416-417) by equal efficiency, neutralization and dosage of GPE- and developed recombinant strain vaccines. Please comment.
L403-404. Please check the provided information concerning leukopenia and thrombocytosis, and provide normal range.
For example, human normal range leukocytes in blood (similar for pigs) is 4-10 x10^9/L. In Fig S5a, S6 Pigs demonstrated leukocytosis - 100 x 10^9/L before inoculation and then doubling at 21 dpi. What is the reason if animals were health? Could leukopenia be assessed in this case?
Platelets human normal range is 4-5 x10^12/L. Pigs showed 4-10 x10^11/L in Fig 5Sb, S6. What is the reason of thrombocytopenia?
Table 5 and Table 6 can be combined since they duplicate information.
Author Response
Response to Reviewer 2:
Thank you for your careful review of our manuscript. The modifications made according to your comments are highlighted in light green.
- The submitted manuscript describes the development and evaluation of a new vaccine candidate against swine fever virus by substitution of the E2 antigen in the currently used GPE- vaccine with the E2 antigen from the wild strain. The study is well done and clearly written. The paper presents evidence of antigenic differences between E2, the major protective antigen of swine fever virus, in the vaccine and the wild strain Mie2019, and the replacement of the antigen with a more relevant one. However, such replacement did not improve the vaccine's efficacy against field strains. Therefore, other pathogenicity factors should be considered and investigated to increase the protective effect. This should be highlighted in the paper.
[Response]
Thank you for your comment.
It is known that the induction of antibodies against E2 protects pigs against highly virulent strains (Reference #30). Therefore, we believe that there is no need to consider other viral proteins.
The reason why the newly produced vGPE–/HiBiT/Mie E2/PAPeV Erns was as effective as the conventional vGPE– in animal experiments is thought to be because we could not plan experiments using newborn piglets with maternal antibodies derived from sows. On the other hand, a study to prepare pigs with maternal antibodies would require significant experimental preparation, including breeding facilities. The fact that the vGPE–/HiBiT/Mie E2/PAPeV Erns was as effective as the vGPE– in this study is a significant result because it will serve as the basis for conducting the next large-scale animal experiment.
[Revision in the text]
The text was modified in Lines 588-595 and 606-609.
- L 21 Please decipher “GPE− vaccine”, so that it is clear, for example, “based on CSFV strain deficient in glycoprotein E”. Avoid complex abbreviations which may be unknown for readers and are difficult to differentiate.
[Response]
Thank you for your comment. The GPE− is a proper noun as a vaccine strain and can be explained as “CSFV that adapted to guinea pig cells and does not exhibit exaltation of Newcastle disease virus phenomenon”. This was clearly stated in the initial description in the introduction section of the main text. On the other hand, in the abstract, this explanation requires too many words; therefore, we have only stated that GPE− is a proper noun as a vaccine strain.
[Revision in the text]
The text was modified in lines 21 and Lines 51-53.
- L28 The reason of PAPeV Erns substitution is not evident. Please, clarify. How Mie protein substitution with proteins from the other viruses provide protectivity against CSFV/Mie?
[Response]
Thank you for your comment. As the reviewer pointed out, we did not explain the role of Erns as an immunogen of vaccine and that it can be replaced with other pestivirus Erns. This explanation has been added to the revised text.
[Revision in the text]
Text was modified in Lines 29-30.
- L132 Specify how long after vaccination of 20 pigs the sera were collected.
[Response]
Thank you for your comment. In this farm, vaccines were administered at 30-40 days of age, and serum samples were collected at 100-120 days of age. This explanation has been added to the revised text.
[Revision in the text]
The text was modified in Lines 134-135.
- L300-306. “The Antigenic Gap” is expressed in Fig 2 quantitatively. This way the authors did not confirm antigen differences between the vaccine and field strains. The obtained results may be due to different expression/presentation and therefore different binding/neutralization of the same antigen in these strains. Provide explanation comments or direct confirmation, differences between their E2 antigens.
[Response]
Thank you for your comment. This experiment was conducted to evaluate the neutralizing activity of vaccine-induced antibodies against the current field isolates. Therefore, no detailed antigenicity analysis was performed. Therefore, the explanation has been revised to align with the findings. For your reference, the amino acid level discussion of these two strains was described in the second paragraph of the discussion part.
[Revision in the text]
The text was modified in Lines 307-310.
- L330 Fix a typo ‘filed strain’
[Response]
Thank you for your comment. This typo was revised according to your comment.
[Revision in the text]
The text was modified in Line 337.
- L333-335. I do not agree with the authors' conclusion from Table 1 results that the recombinant viruses are antigenically similar to the field strain Mie/2019. The results demonstrate that the neutralizing activity of the anti-vaccine virus sera is equal to that of the anti-recombinant virus sera (76=76). Thus, the E2 substitution is ineffective. This is also confirmed (L416-417) by equal efficiency, neutralization and dosage of GPE- and developed recombinant strain vaccines. Please comment.
[Response]
Thank you for your comment. We apologize for the inadequate explanation of the results presented in Table 1 and misunderstanding by the reviewer. As the revised description indicates, the reactivity of serum to these four viruses was dependent on the origin of the E2 gene, and this result confirms our argument.
I do not think that the fact that the vaccination dosage of vGPE–/HiBiT/Mie E2/PAPeV Erns and vGPE– is are same, as stated in L416-417, is directly related to the efficacy of vGPE–/HiBiT/Mie E2/PAPeV Erns as a vaccine. In terms of the challenge study, I am responding in the comment #1.
[Revision in the text]
The text was modified in Lines 335-342 and Table 1.
- L403-404. Please check the provided information concerning leukopenia and thrombocytosis, and provide normal range.
[Response]
Thank you for your comment. First, there was a serious mistake in the text: leukopenia and thrombocytopenia are induced by CSFV infection. Information about leukopenia (≤8 x 103/µL, normal range in adult: 12–30 x 103/µL) and thrombocytopenia (≤10 x 104/µL, normal range in adult: 25–60 x 104/µL) was included in the text. In the explanation given in Figure S4, it is necessary to explain the definitions of these two words, so this explanation has been added to that section.
[Revision in the text]
The test was modified in Lines 383-386 and Lines 416.
- For example, human normal range leukocytes in blood (similar for pigs) is 4-10 x10^9/L. In Fig S5a, S6 Pigs demonstrated leukocytosis - 100 x 10^9/L before inoculation and then doubling at 21 dpi. What is the reason if animals were health? Could leukopenia be assessed in this case?
[Response]
Thank you for your comment. As noted above, the normal white blood cell count in adult pigs is 12–30 x 103/µL. On the other hand, leukocyte counts in young pigs, as in humans, are higher than normal. In other words, the numbers in Figs. S4, S5, and S6 that are suspected of leukocytosis are considered to be within the normal range for 2-week-old piglets after weaning. The same explanation can be provided for the platelet counts.
Furthermore, an increase in leukocyte counts after vaccination is a biological response to immunization with live vaccines, a phenomenon that has been reproduced in the previous study. Therefore, we do not believe that these phenomena are the adverse effects of vaccination.
Text explaining these points has been added to the revised text.
[Revision in the text]
The test was modified in Lines 387-389 and Lines 416-418.
10 Platelets human normal range is 4-5 x10^12/L. Pigs showed 4-10 x10^11/L in Fig 5Sb, S6. What is the reason of thrombocytopenia?
[Response]
Thank you for your comment. As answered by suggestions #8 and #9, the normal range of platelets in adult pigs is 25–60 x 104/µL). This number translates to 2.5–6.0 x 1011 per liter. In other words, the platelet counts in Figures S5 and S6, as noted by the reviewers (4-10 x10^11/L), were within the normal range. We suspect that the number highlighted by the reviewer (4-5 x10^12/L in humans) may be incorrect.
[Revision in the text]
This has already been corrected in the above question, and there are no new modifications.
- Table 5 and Table 6 can be combined since they duplicate information.
[Response]
Thank you for your comment. We have also considered incorporating your opinion. However, combining the two tables into one would make it difficult to understand the two important messages we would like to convey.
Table 5: Neutralizing antibody titers before and after vaccination and after challenge with the Mie/2019 strain
Pigs did not have neutralizing antibodies against CSFV before the animal experiment.
Neutralizing antibodies induced by vaccination were not detected at 0 dpc.
The properties of neutralizing antibodies detected at the end of the challenge test (14 dpc) differed.
Table 6: Results of antibody detection using the IC test strip and supplementary information for interpretation. (implementation record of vaccination and virus challenge, correlation between neutralizing antibody titers and antibody detection with IC test strip)
We removed the information on infectivity titers from Table 6 and maintained these results as two tables. We hope that you will understand our explanation.
[Revision in the text]
A part of Table 6 was modified in Lines 515-516.
Reviewer 3 Report
Comments and Suggestions for Authors
I reviewed the manuscript entitled “Generation of vaccine candidate strains that antigenically match classical swine fever virus field strain”. Based on serologic evidence, determining that antibody response produced by a live attenuated vaccine used in Japan is not efficient to neutralize current CSFV circulating strains, authors developed a new vaccine candidate by swapping structural proteins Erns and E2 glycoprotein.
Overall, I think there is an important flaw in the experimental design of this study. First, authors show in the results the lack of efficiency of the current vaccine to elicit an efficient neutralizing antibody response against a current circulating CSFV strain in Japan. However, since pigs were challenged in absence of antibodies (early during the vaccination), the experimental design used by the authors did not assess the role of antibodies produced by the new vaccine candidate in the protection of CSFV, which was the main argument about the necessity of producing a new vaccine. In terms of protection, there is no difference in the efficacy between both vaccines.
In my opinion, this study should be restructured to show the ability of the current vaccine to protect against the challenge with a WT CSFV from a different genotype, even when vaccine matching studies show the lack of efficacy of a vaccine strain to neutralize a viral strain from a different genotype.
Author Response
Response to Reviewer 3:
Thank you for your careful review of our manuscript. The modifications made according to your comments are highlighted in light green.
- Overall, I think there is an important flaw in the experimental design of this study. First, authors show in the results the lack of efficiency of the current vaccine to elicit an efficient neutralizing antibody response against a current circulating CSFV strain in Japan. However, since pigs were challenged in absence of antibodies (early during the vaccination), the experimental design used by the authors did not assess the role of antibodies produced by the new vaccine candidate in the protection of CSFV, which was the main argument about the necessity of producing a new vaccine. In terms of protection, there is no difference in the efficacy between both vaccines.
[Response]
Thank you for your comment. The reason why the newly produced vGPE–/HiBiT/Mie E2/PAPeV Erns was only as effective as the conventional vGPE– in animal experiments is thought to be because we could not plan experiments using newborn piglets with maternal antibodies derived from sows. On the other hand, a study to prepare pigs with maternal antibodies would require significant experimental preparation, including breeding facilities. The fact that the vGPE–/HiBiT/Mie E2/PAPeV Erns was as effective as the vGPE– in this study is a significant result because it will serve as the basis for conducting the next large-scale animal experiment.
In summary, we have added a detailed description of the limitations of the current manuscript, particularly the significance of the present and future studies.
[Revision in the text]
The text was modified in Lines 588-595 and 606-609.
- In my opinion, this study should be restructured to show the ability of the current vaccine to protect against the challenge with a WT CSFV from a different genotype, even when vaccine matching studies show the lack of efficacy of a vaccine strain to neutralize a viral strain from a different genotype.
[Response]
Thank you for your comment. Generally, CSFV strains are categorized into three major genotypes. Cross-reactive neutralizing antibodies inhibit the growth of CSFV genotype 2 in pigs vaccinated with CSFV genotype 1 (Reference #30). In addition, limited replication of the challenge virus may be achieved partially by cellular-mediated immunity elicited by the live-attenuated vaccine (Reference #10 and #31). If pigs immunized with the vGPE–/HiBiT/Mie E2/PAPeV Erns were challenged with CSFV of different genotypes (genotypes 1 and 3), its efficacy as a vaccine would also be confirmed. We believe that this study is necessary in the future, but as mentioned above, we would like to conduct a study using piglets with only maternal antibodies to demonstrate that vGPE–/HiBiT/Mie E2/PAPeV Erns is more effective than vGPE– against the challenge of CSFV genotype 2.
[Revision in the text]
The text was modified in Lines 588-595 and 606-609.
Round 2
Reviewer 2 Report
Comments and Suggestions for Authors
The reviewer's recommendations have been taken into account and the necessary explanations and comments have been made.
Reviewer 3 Report
Comments and Suggestions for Authors
I like to thank the authors for their responses to my comments. I consider that the content of the manuscript was improved. At this point, I don't have more concerns about it.